# Controlled Release of Encapsuled Stromal-Derived Factor 1α Improves Bone Marrow Mesenchymal Stromal Cells Migration

**DOI:** 10.3390/bioengineering9120754

**Published:** 2022-12-02

**Authors:** Karolina Bajdak-Rusinek, Agnieszka Fus-Kujawa, Katarzyna Jelonek, Monika Musiał-Kulik, Piotr Paweł Buszman, Dorota Żyła-Uklejewicz, Adrianna Walentyna Sekowska, Janusz Kasperczyk, Paweł Eugeniusz Buszman

**Affiliations:** 1Department of Medical Genetics, Faculty of Medical Sciences in Katowice, Medical University of Silesia, Medykow 18 Street, 40-752 Katowice, Poland; 2Center of Polymer and Carbon Materials, Polish Academy of Sciences, M. Curie-Sklodowskiej 34, 41-819 Zabrze, Poland; 3Center for Cardiovascular Research and Development, American Heart of Poland, 40-028 Katowice, Poland; 4Cardiology Department, Andrzej Frycz Modrzewski Krakow University, 30-705 Kraków, Poland; 5Students Scientific Society, Faculty of Medical Sciences in Katowice, Medical University of Silesia, 40-752 Katowice, Poland; 6Department of Epidemiology, Medical University of Silesia, 40-752 Katowice, Poland

**Keywords:** SDF-1α, bone marrow mesenchymal stromal cells (bmMSCs), biodegradable polymeric microspheres, cell migration

## Abstract

Stem cell treatment is a promising method of therapy for the group of patients whose conventional options for treatment have been limited or rejected. Stem cells have the potential to repair, replace, restore and regenerate cells. Moreover, their proliferation level is high. Owing to these features, they can be used in the treatment of numerous diseases, such as cancer, lung diseases or ischemic heart diseases. In recent years, stem cell therapy has greatly developed, shedding light on stromal-derived factor 1α (SDF-1α). SDF-1α is a mobilizing chemokine for application of endogenous stem cells to injury sites. Unfortunately, SDF-1α presented short-term results in stem cell treatment trials. Considering the tremendous benefits of this therapy, we developed biodegradable polymeric microspheres for the release of SDF-1α in a controlled and long-lasting manner. The microspheres were designed from poly(L-lactide/glycolide/trimethylene carbonate) (PLA/GA/TMC). The effect of controlled release of SDF-1α from microspheres was investigated on the migration level of bone marrow Mesenchymal Stromal Cells (bmMSCs) derived from a pig. The study showed that SDF-1α, released from the microspheres, is more efficient at attracting bmMSCs than SDF-1α alone. This may enable the controlled delivery of selected and labeled MSCs to the destination in the future.

## 1. Introduction

Stem cell therapy is currently the most promising approach for a range of medical conditions and diseases for which few treatments have been developed. Stem cells are the basic cells that can differentiate into specialized cell types of tissues or organs in the body. They can develop into blood, brain, bones, and all organs of the body [1]. Because of these traits, they have the potential to repair, restore, replace and regenerate cells, and therefore can be used in the treatment of many different diseases, such as cancer [2], lung diseases [3] or ischemic heart diseases [4].

The benefits of stem cell therapy are enormous. Importantly, they directly affect the increase in the number of living cells at the site of damage due to high proliferation levels. Moreover, they improve the survival and functioning of the existing tissue thanks to secreted factors. Unfortunately, despite the huge progress that has been made in this field, stem cell therapies still fail to fulfill their role in disease treatment. This is due to the poor survival of the administered cells, as well as poor implantation efficiency of these cells after stem cell transplantation, which was observed for example in the case of bone marrow mononuclear cells in the treatment of myocardial infarction (MI) [5]. It is estimated that, following stem cell implantation to treat tissue damage from a heart attack, less than 10% is retained at the injection site. Moreover, such a small number of surviving cells is insufficient to obtain a therapeutic effect [6].

Although clinical trials in patients with cardiovascular diseases have shown that cellular therapies are generally safe, they have poor efficacy in improving heart function. It was found that the transplanted cells do not receive the appropriate signals to recruit properly and implant into the host tissue [7]. One of the main signals for attracting stem cells is stromal-derived factor 1α (SDF-1α) [8]. In recent years, the SDF-1α chemokine has been proposed as a candidate for the mobilization of endogenous stem cells to populate the ischemic heart, increase capillary density and finally improve heart function. Liu et al. showed that local injection of SDF-1α into the host tissue, followed by the administration of stem cells, improves the colonization and implantation of stem cells. Unfortunately, although direct injection of SDF-1α showed some benefits in cell therapy, its effects were short-term [9]. Genetic modification of bmMSCs in order to obtain SDF-1α overexpression can increase the colonization of stem cells and improve cardiac function in animal models [10,11]. However, this kind of modification may pose a safety risk as it may result in a sustained increase in SDF-1α expression with unknown consequences.

Therefore, the challenge is to identify an appropriate delivery system for both the amount as well as the regulation of the duration of SDF-1α expression to mobilize endogenous stem cells to colonize the desired sites. In recent years, the controlled release of drugs based on biomaterials has been proposed as a potential source of delivering therapeutics in a sustained, transient and controlled manner.

Due to the long history of safe in vivo use and use in clinically licensed products, biodegradable microspheres, in particular poly (lactide/glycolide), are the most attractive material for this purpose. Polylactide microspheres and PLGA (poly (lactic/glycolic acid)) have previously been studied as vehicles for the delivery of vaccine antigens in the form of peptides, recombinant proteins, inactivated viruses or DNA preparations. The results showed that they increased both antibody production and T-cell immune responses in vivo [12].

In this article, we investigate the effect of controlled release of SDF-1α from microspheres on the migration level of bmMSC cells. Our idea is based on the development of biodegradable polymeric microspheres for the release of SDF-1α in a controlled and long-lasting manner.

## 2. Materials and Methods

### 2.1. Animal Studies

The study was performed in the Center for Cardiovascular Research and Development of American Heart of Poland. All procedures were approved by the Animal Ethics Committee (Contract No. 64/2018, 19/2021). All animals received standard care outlined in the study protocol and in accordance with the Animal Welfare Act and the “Guide for the Care and Use of Laboratory Animals.” A total of 21 domestic swine, with the mean weight of 40 kg, were enrolled in this study.

### 2.2. Bone Marrow MSCs Isolation

A porcine bone marrow aspiration biopsy was performed under aseptic conditions. The aspirate was collected into the heparin (5000 IU/mL) (Polfa S.A., Warsaw, Poland) and thoroughly mixed to prevent clots. The homogeneous bone marrow suspension was then diluted in the ratio 1:1 in Dulbecco’s Phosphate Buffered Saline *w*/*o* calcium, magnesium (DPBS) (PAN Biotech, GmbH, Aidenbach, Germany) and layered carefully on Ficoll-PagueTMPlus (density 1.077 ± 0.001 g/mL) (Cytivia, Shrewsbury, MA, USA). Cell suspension was centrifuged at 500× *g* for 30 min. Subsequently, the mononuclear cell layer was collected and centrifuged again. Afterwards, the supernatant was aspirated and the pellet was resuspended in DPBS. The isolation efficiency was calculated by determining the cell density and viability using Automated Cell Counter TC20 (Bio-Rad, Hercules, CA, USA).

### 2.3. Cell Culture

Bone marrow MSCs were seeded at density 5.0 × 10^6^ cells in a cell culture medium consisting of Mesenchymal Stem Cell Basal Medium (ATCC, Manassas, VI, USA) supplemented with Mesenchymal Stem Cell Grow Kit—low serum (ATCC, Manassas, VI, USA), 10,000 U/mL Penicillin, 10 mg/mL streptomycin, 25 ug/mL amphotericin B (PAN Biotech GmbH, Aidenbach, Germany) at 37 °C and 5% CO_2_.

### 2.4. Adhesion and Proliferation

Bone marrow MSCs were seeded at a density 5 × 10^4^ cells per each well. Adhesion assays were performed after 2.5, 4 and 8 h and proliferation assays were performed after 24, 48, and 72 h of cell culture. After each time point, the cells number was assessed using the Alamar blue reagent as a 10% solution with cell culture medium, for which the whole procedure has been previously described [13].

### 2.5. Flow Cytometry

The following monoclonal antibodies (Mouse anti-Human) were used for flow cytometric analysis of bmMSCs: CD73 FITC, CD90 FITC, CD105 APC (STEMCELL Technologies, Cambridge, MA, USA). CD34 was used as a negative control with Goat anti-Mouse IgG (H + L) Highly Cross-Adsorbed Secondary Antibody, Alexa Fluor 488 (ThermoScientific, Waltham, MA, USA). BmMSCs were stained according to the manufacturer’s instructions as described before [14]. Unstained bmMSCs were used as a control. Analysis was performed on FACSAria I Flow Cytometer (BD Biosciences, Haryana, India).

### 2.6. SDF-1α Cytotoxicity Assay

BmMSCs cells were seeded in 24-well plates at a density of 5 × 104 per well. The following day, the cell culture medium was supplemented with human stromal-derived factor 1 alpha (SDF-1α) to the desired final concentrations: 0 (control), 50, 100, 150, 200, and 250 ng/mL. The cells were incubated for an additional 24 h, then cells were rinsed with pre-warmed DPBS (ThermoScientific, Waltham, MA, USA). The cells viability was evaluated with the Alamar blue reagent. After the addition of 200 µL of prewarmed Alamar blue reagent in a cell culture medium at a final concentration 10% in each well, the cells were incubated under standard conditions at 37 °C and 5% CO_2_ for 1 h. Subsequently, 100 µL of the mixture from each well was transferred to a new well of a 96-well TPPTM plate (PerkinElmer, Waltham, MA, USA), and an intensity of the fluorescence emission was detected at 590 nm using a VICTOR^TM^ Multilabel Plate Reader (PerkinElmer, Waltham, MA, USA) with a 560 nm excitation source. The cell viability was assessed based on the percent of living cells compared to the control cells untreated with SDF-1α.

### 2.7. MSC-Derived Osteoblasts

For osteogenic induction of MSCs, the cells were seeded as previously described [15]. When cells reached 90% confluency, growth media were replaced with MesenCult™ Osteogenic Differentiation Kit (Human) (StemCells Technologies, Cambridge, MA, USA) according to the manufacturer’s instructions.

After 14 days of differentiation cells were fixed with 4% paraformaldehyde for 30 min at RT, washed three times with demineralized water and stained with an Alizarin Red S solution (Sigma Aldrich, Darmstadt, Germany) for 20 min at RT. Afterwards, the cells were washed three times with DPBS (ThermoScientific, Waltham, MA, USA).

### 2.8. MSC-Derived Adipocytes

For adipogenic induction of bmMSCs the cells were seeded as previously described [15]. When cells reached 90% confluency, growth media was replaced with MesenCult™ Adipogenic Differentiation Kit (Human) (StemCells Technologies, Cambridge, MA, USA) according to the manufacturer’s instructions.

On the 35th day of differentiation, the cells were fixed with 4% paraformaldehyde for 30 min at RT, washed three times with demineralized water and stained with Oil Red O (Sigma Aldrich, Darmstadt, Germany) as previously described [16].

### 2.9. MSC-Derived Chondrocytes

For chondrogenic induction of bmMSCs, the cells were seeded in MesenCult™-ACF Chondrogenic Differentiation Kit, as previously described [17]. When cells reached 90% confluence, growth media was replaced with MesenCult™ Adipogenic Differentiation Kit (Human) (StemCells Technologies, Cambridge, MA, USA) according to the manufacturer’s instructions.

On 21st day of differentiation, the cells were fixed with 4% paraformaldehyde for 30 min at RT, washed three times with demineralized water and incubated with an 60% isopropanol (POCH, Gliwice, Poland) for 5 min at RT. Afterwards, the cells were stained with Alcian Blue (ScienceCell, Carlsbad, CA, USA) for 20 min at RT and washed 3–5 times with DPBS (ThermoScientific, Waltham, MA, USA).

### 2.10. Preparation of the Microspheres with SDF-1α

The microspheres (MS) were prepared from poly(L-lactide/glycolide/trimethylene carbonate) (PLA/GA/TMC) synthesized at the Centre of Polymer and Carbon Materials, Polish Academy of Sciences. The number-average molar mass (Mn) of the terpolymer was 44 kDa and the comonomer unit ratio of lactide:glycolide:TMC was 76:10:14. The microspheres were obtained using a water/oil/water (*w*/*o*/*w*) emulsion method. Briefly, 6 mg of SDF-1α was mixed with 49 mg of bovine serum albumin (BSA) and dissolved in 1 mL of deionized H_2_O. The oil phase consisted of 250 mg of polymer dissolved in the 2 mL of dichloromethane (CH_2_Cl_2_). The aqueous solution of SDF-1α with BSA was added to the polymer in CH_2_Cl_2_ and sonicated for 15 s (Hielscher UP200Ht). The *w*/*o* phase was added dropwise to 200 mL of 5% PVA (for 90 s), which served as an aqueous phase, and was further emulsified by a homogenizer (Kinematica, Polytron PT 2500 E) at 13,000 rpm for another 60 s. The resulting emulsion was stirred on a magnetic stirrer at a rate of 100 rpm at ambient temperature to ensure solvent evaporation (overnight). The prepared MS were collected by centrifugation at 2500 rpm (Eppendorf 5810R) for 15 min at 20 °C. The precipitate was washed with distilled water and centrifuged. This step was repeated three times. The obtained MS were then freeze-dried (Christ, Alpha 1-2 LD plus) and stored at 4 °C.

### 2.11. Characterization of the MS

#### 2.11.1. Microscopic Observation

The surface of the lyophilized MS was characterized with the use of a scanning electron microscope (SEM; FEI Company, Quanta 250 FEG) operating under a low vacuum of 80 Pa. Samples were fixed to a holder plate by the use of conductive carbon tape. Measurements were carried out at an electron accelerating voltage of 5 kV, without sputtering with gold on the material surface. ImageJ 1.45 s software was employed to determine the size of the MS as well as the size distribution.

#### 2.11.2. Encapsulation Efficiency

The percentage of SDF-1α encapsulated into MS (EE) was evaluated using the below formula:EE (%) = W_SDF_/W_SDF0_ × 100%
where: W_SDF_ is the weight of SDF-1α in the MS, and W_SDF0_ is the weight of SDF-1α taken to prepare the MS. To determine the SDF-1α entrapped in MS, the extraction method was adopted [18,19]. Briefly, 1 mg of the MS was added to 0.5 mL of dichloromethane and stirred on magnetic stirrer at 500 rpm for 30 min, then 4.5 mL of PBS was added, and the solution was mixed on vortex for 30 min. The quantitative analysis of SDF-1α was conducted on the basis of the ELISA assay (Human SDF-1 ELISA Kit, AVIVA Systems Biology). The procedure was repeated three times.

#### 2.11.3. In Vitro Release of the SDF-1α

The suspension of 5 mg MS with SDF-1α in 5 mL of phosphate-buffered saline (pH 7.4) was placed in 15 mL tubes and incubated at 37 °C under constant shaking of 240 rpm. The buffer was renewed each week. For this purpose, after samples centrifugation (9000 rpm for 15 min at 20 °C), the supernatants were replaced with fresh medium. At the specified time points (1 day, 2 days, 3 days, 7 days, 14 days, 21 days), after the centrifugation of samples the precipitate (MS) was stored at −80 °C for subsequent analysis of the SDF-1α residues. Accordingly, the extraction of chemokine from MS described in the section above (Encapsulation efficiency) was used. The quantitative analysis of SDF-1α was conducted on the basis of the ELISA assay (Human SDF-1 ELISA Kit, AVIVA Systems Biology). The procedure was repeated three times.

### 2.12. Transwell Migration Assay

The migration assay was designed using Transwell plates (Corning Costar, Cambridge, MA, USA) with 8 µm pore filters. In the upper chamber were loaded 0.5 × 10^6^ cells, which were suspended in a serum-free medium. Cells were allowed to migrate toward medium in the lower chamber filled with the same medium containing 2% FBS. For SDF-1α treatment, SDF-1α was added at concentrations of 0 (control), 50, 100, 150, 200 and 250 ng/mL and microspheres releasing SDF-1α (200 ng/mL). To avoid the burst effect the microspheres were pre-incubated for 48 h in PBS. After 24 h of incubation at 37 °C and 5% CO_2_, non-migratory cells were carefully removed from upper face of the Transwell insert with a cotton swab. The cells that migrated to the lower side of the filter were washed with pre-warmed PBS (Sigma Aldrich, Darmstadt, Germany). Subsequently, 200 µL of pre-warmed Alamar blue reagent in cell culture medium was added to a final concentration of 10%. Cells were incubated at 37 °C and 5% CO_2_ for 1 h. Subsequently, 100 µL of the mixture was transferred to a new well of 96-well TPP plate (PerkinElmer, Waltham, MA, USA) and the fluorescence emission was monitored at 590 nm using a VICTORTM Multilabel Plate Reader (PerkinElmer, Waltham, MA, USA) with a 560 nm excitation source. Cell density and viability were assessed based on the percentage of live cells that migrated through the membrane when compared to the control that was not treated with chemoattractant.

### 2.13. Statistical Analysis

Statistical analyses were performed with Microsoft Excel software. Normalized relative expression levels were used to calculate the mean and the SD of all experiments (represented by columns and error bars in the figures). The two-tailed equal variance Student’s *t* test was used to access statistical significances. Multiple group comparisons were performed using one-way ANOVA followed by Tukey’s post hoc test to evaluate the differences between the groups. In all figures, *p* values of statistical significance are represented as follows: * *p* < 0.05; ** *p* < 0.01.

## 3. Results

### 3.1. Isolation, Expansion and Characterization of Porcine Bone Marrow MSCs

MSCs were successfully isolated from the bone marrow of domestic pigs. After adhesion was observed in three time points, bmMSCs gained round shapes. When the medium was changed the majority of non-adherent cells, such as red cells, were eliminated. After 24 h of cells culture, the adherent cells extended to become spindle-shaped and gradually proliferated (Figure 1A,B).

We confirmed the immunophenotype of obtained pig bmMSCs using flow cytometry. Our results revealed that the established cells were strongly positive for MSCs markers CD73, CD90 and CD105 and negative for the endothelial cell marker CD34. The isotype control was negative (Figure 2).

Overall, our results clearly showed that the cells we obtained were bmMSCs. Such a comprehensive analysis is crucial in terms of further regenerative analysis.

### 3.2. Bone Marrow MSCs Demonstrate Osteogenic, Adipogenic and Chondrogenic Differentiation Potential

To further assess the multipotent differentiation capacity of bmMSCs, cells were cultured with osteogenic, chondrogenic and adipogenic media to obtain an osteoblast, chondrocyte and adipocyte phenotypes respectively.

After 14 days of cells differentiation in osteogenic media, calcification deposits of cells were detected with cells staining positive for Alizarin Red S. In turn, chondrogenic differentiation of bmMSCs was confirmed by Alcian Blue staining for matrix synthesis of glycosaminoglycans (GAGs), something that is an important extracellular matrix (ECM) component of the cartilage tissue. After 3 weeks we observed intensely blue color indicative of cartilage extracellular matrix.

Finally, cells treated with adipogenic medium showed an adipocyte phenotype after 35 days with the appearance of cytoplasmic lipid vacuoles detected by Oil Red O staining. Control cells showed no adipocyte formation.

Overall, the results showed that the bmMSCs we isolated have the potential to differentiate into osteoblasts, chondrocytes and adipocytes (Figure 3). The functionality of the swine bmMSCs is crucial for their use in regeneration applications.

### 3.3. Effect of SDF-1α on Cells Viability

In order to use cytokines released from biodegradable microspheres to attract MSCs for the treatment of ischemic cardiomyopathy, the first step is to assess whether the SDF-1α is not toxic for tested cells. For this purpose, we performed a cytotoxicity test where we treated bmMSC with the SDF-1α in the range of concentration 50–250 ng/mL (Figure 4). The results showed that none of the concentrations used are toxic to bone marrow MSCs, something that allows their successfully application to further stages of research.

### 3.4. Effects of SDF-1α on bmMSCs Migration In Vitro

In order to confirm the activity of SDF-1α as a chemoattractant for bmMSCs, we performed a migration test. A Transwell-based migration assay was established to quantitatively evaluate bmMSCs migration in vitro. As shown in Figure 5, compared with the untreated control group, the average number of migrated cells increased significantly with 50, 100, 150, 200 and 250 ng/mL SDF-1α induction, which reached a peak at 200 ng/mL.

Our results revealed that SDF-1α induced bmMSCs migration in a dose-dependent manner, and maximum migration ratio was observed following treatment with SDF-1α at 200 ng/mL, which indicates that SDF-1α is important in bmMSCs migration. The use of SDF-1α will allow researchers to increase the percentage of migrating cells, thus creating hope for increasing the efficiency of bmMSCs delivery to destination sites.

### 3.5. Encapsulation and Release of SDF-1α from MS

To experimentally test controlled release of chemoattractants, we produced microspheres containing SDF-1α.

The microspheres (MS), loaded with SDF-1α, were prepared from poly (L-lactide/glycolide/trimethylene carbonate) (PLA/GA/TMC). The encapsulation efficacy of the SDF-1α was 67%, so the final content of SDF-1α in microspheres was 1.4% (*w*/*w*). The SEM analysis confirmed a regular spherical shape of the MS with a smooth surface (Figure 6). The size of the MS was in a range of 3–12 μm. The in vitro release of SDF-1α was conducted at 37 °C and the immunoenzymatic method was applied for quantitative evaluation. The release profile of the SDF-1α is presented in Figure 7. A significant amount of SDF-1α was released already after 1 day of incubation, which is typical for this kind of delivery system. This phenomenon may be considered advantageous because it provides high initial dose of the active agent. The release of SDF-1α proceeded in a regular way (21%, 28%, 32% released SDF after 1, 7 and 14 days, respectively), and so finally 37% of chemokine was released after 21 days.

### 3.6. Released SDF-1α from the Microsphere Induces In Vitro Migration of bmMSCs

The ability of SDF-1α which have been released from microspheres to mobilize bmMSCs in vitro was evaluated using a Transwell migration assay. As shown on Figure 8, cell migration across the Transwell membrane was identified by treatment with SDF-1α. The numbers of recruited bmMSCs in the groups that contained SDF-1α released from the microspheres were significantly increased compared to using SDF-1α alone. Nevertheless, in both cases these values were higher than in the control.

The obtained results show that the SDF-1α released from the microspheres is even more efficient at attracting bmMSC than standard SDF-1α. Moreover, once released from the microspheres, SDF-1 is able to attract cells, which proves that our encapsulation method preserves the protein’s bioactivity, and the chosen method of microspheres preparation does not denature it.

## 4. Discussion

Stromal cell-derived factor1 (SDF-1) is a chemokine produced by bone marrow stromal cells. Its most important function includes promoting the stem cell homing and migration [20]. In 2003, Askari et.al. identified SDF-1 as a first myocardial homing factor [21], which is necessary for the mobilization of endogenous stem cells homing to the ischemic heart. Since stem cells are involved in the natural response to ischemic tissue damage, they have become a promising target for clinical trials to repair and replace damaged myocardium [22]. Indeed, in cell therapy trials of myocardial infarction (MI) and heart failure, bone marrow Mesenchymal Stem Cells (bmMSCs) dominate. The studies conducted so far have shown that the use of these cells in cell therapy is safe and leads to an improvement in cardiac function, as well as in clinical outcomes, such as MI relapses [23,24,25].

BmMSCs are specialized adult bone marrow stem cells that have a characteristically potent ability to differentiate. In the present study, bmMSCs were obtained from porcine material and cultured. Mesenchymal stem cells are the most frequently used cell type for application in regenerative medicine [26]. Numerous studies have shown the valuable effects of MSC-based therapies for the treatment of pathologies. The crucial thing is to assess their viability, proliferation rate and functionality after isolation from animal sources. MSCs are attractive candidates for biological cell-based tissue repair approaches because of their extensive proliferative ability in culture. Our findings show that cells adhere to a culture surface within 8 h and that there is a progress of proliferation within a time (Figure 1A,B). It has been also proven that MSCs from animal sources display the expression of specific markers. This is in agreement with our results where we confirmed MSCs’ phenotype with markers specific to this cells: CD73, CD90 and CD105 (Figure 2). For clinical application, it is necessary to obtain as a high percentage of positive cells as it is possible. These cells should also be a homogeneous mixture.

Before isolated cells will be used for further applications such as clinical trials, their functionality and ability to differentiate should be confirmed. Chondrogenesis, adipogenesis and osteogenesis are well-orchestrated processes mediated by various interactions at a molecular level. The molecular regulation of MSC’s differentiation has been comprehensively studied. Regulation of MSC differentiation would be crucial in the design of three-dimensional culture systems and targeted delivery of bmMSCs [27]. We confirmed the ability of isolated MSCs to differentiate into osteoblasts, chondrocytes and adipocytes (Figure 3).

Importantly, after an acute myocardial infarction, SDF-1α levels increase. Thanks to this, SDF-1α can mobilize stem cells in homing to the ischemic heart and improve heart function. Unfortunately, this effect is short-term [28]. Conversely, in the absence of damage, the amount of SDF-1α is insufficient to induce recruitment of stem cells into the ischemic myocardium [29]. Therefore, the administration of SDF-1α after MI results in increased recruitment of stem cells [30]. Our in vitro tests demonstrated a positive correlation between bmMSCs migration and SDF-1α concentration. Maximum migration was observed when the SDF-1α concentration was 200 ng/mL (Figure 5).

Since the short-term increase in SDF-1α after MI is not sufficient to repair the damaged tissue, sustained expression of SDF-1α is needed for this purpose.

To date, various attempts have been made to increase the concentration of SDF-1α. Among others, multiple injections of SDF-1α were used, but unfortunately it diffused quite quickly [31,32]. Another proposal was to transplant cells that overexpress the SDF-1 gene. Unfortunately, the use of gene therapies is difficult due to poor dose and duration control. Moreover, this method is limited due to the low efficiency of gene transfer and the risk of genomic integration, the latter of which is associated with the risk of tumorigenesis [33]. Therefore, this method has no therapeutic application yet.

That is why it is so important to create an appropriate system of controlled SDF-1α release in order to mobilize the migration of endogenous stem cells to the desired destination.

Microspheres have been selected as a delivery system of the SDF-1α. Various kinds of active agents such as proteins or peptides are incorporated in the microspheres to enhance their bioavailability and maintain therapeutic concentrations for long periods of time (days or months) [34]. Most of the reported microspheres so far for delivery of chemokines have been obtained from commercially available copolymers, e.g., poly (lactide/glycolide) (PLGA) [35]. The (PLGA), a US FDA-approved polymer, has been the most widely explored and is gaining more and more interest as a polymer for drug delivery applications due to its advantageous properties, including its biodegradability, non-toxicity, non-immunogenicity, and biocompatibility, but also mechanical strength and processability [36]. The presented microspheres were obtained from poly (L-lactide/glycolide/trimethylene carbonate), which has many advantages in pharmaceutical and biomedical applications. The terpolymer has been synthesized by using zirconium (IV) acetylacetonate [Zr(acac)4] which served as a non-toxic initiator of ROP (ring-opening polymerization) [37,38,39] to provide high biocompatibility. Generally, copolymerization has been broadly used to obtain desired properties in the final materials [40,41] and terpolymers possess even greater potential to tailor properties of the final material compared to copolymers. Addition of another comonomer (TMC) to PLGA may be advantageous, because poly(trimethylene carbonate) undergoes surface erosion without generating acidic by-products. This is something that could increase bio-compatibility as well as protect the labile molecules of the active agent. This is especially important for the administration of microspheres to the isolated sites of organism (e.g., pericardial) where diffusion of the acidic degradation products may be delayed and increase the risk of inflammation reaction. Thus, the selection of poly(L-lactide/glycolide/TMC) for preparation of microspheres loaded with SDF-1α was a novel approach and enabled us to obtain regular spherical shape of the microparticles with smooth surface (Figure 6). Another advantage of the microspheres prepared from poly(L-lactide/glycolide/TMC) is high encapsulation efficiency of the SDF-1α (67%), which resulted in much higher content of the chemokine in the microspheres (1.4% *w*/*w*) compared to the microparticles reported so far [35]. Thus, a lower dose of microspheres may provide high levels of SDF-1α. Moreover, the microspheres with high loading content of the chemokine provided long-term release of the SDF-1α, because only 40% of SDF-1α was released within 21 days (Figure 7). The results published so far from an in vitro SDF-1α release report mostly much faster elution from delivery systems [42,43,44,45]. It is quite certain that such a long release of SDF-1α will be sufficient to induce sustained and beneficial cell migration in various disease like myocardial infarction. This will especially apply to the heart which, after the infarction itself, overexpresses SDF-1α for 7 days after the injury [30].

Typically for microparticulate delivery systems, an increased initial SDF-1α release was observed after 1 day (21%). This effect is explained by the rapid liberation of proteins located adjacent to the microspheres’ surface [46]. The initial more rapid release may be useful in establishing higher local concentration of SDF-1α and providing immediate induction of chemotaxis of the stem cells. This stage was followed by a slower release caused probably by the diffusion of proteins from deeper parts of the microspheres.

Indeed, our studies have shown that SDF-1α, released from microspheres, stimulates the stem cells migration (Figure 8). It also confirms that the encapsulation method preserves the protein’s bioactivity, and the selected preparation method of microspheres (water-in-oil-in-water (*w*/*o*/*w*) emulsion) did not cause the protein denaturation. The preservation of protein bioactivity throughout formulation processes is imperative as denatured proteins tend to be more immunogenic than their native forms [46].

To date, the ideal time for sufficient SDF-1α release to heal the damaged heart is not known. In animals and humans, it takes at least weeks or even months. Therefore, our microspheres, which are capable of controlled and sustained SDF-1α release, may in the future provide clinical benefits to MI patients by extending the duration of action of SDF-1α at the injection site.

More studies are needed to adjust the release profile and evaluate it in vivo. In the future, this could have a clinically significant impact on repairing the heart damage with the use of stem cells. This will allow for controlled delivery of selected and labeled MSCs to the destination.

## 5. Conclusions

This section is not mandatory but can be added to the manuscript if the discussion is unusually long or complex.

## Figures and Tables

**Figure 1 bioengineering-09-00754-f001:**
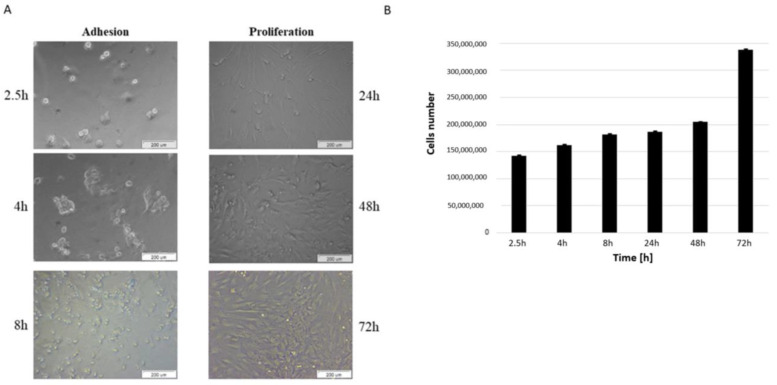
Adhesion and proliferation of bmMSC cells. (**A**) Bright-field images for bmMSCs after 2.5, 4, 8, 24, 48 and 72 h of culture. Scale bar: 200 μm. (**B**) Cells number after 2.5, 4, 8, 24, 48 and 72 h of culture were determined by automated counting. The bars represent the means ± SD (*n* = 3).

**Figure 2 bioengineering-09-00754-f002:**
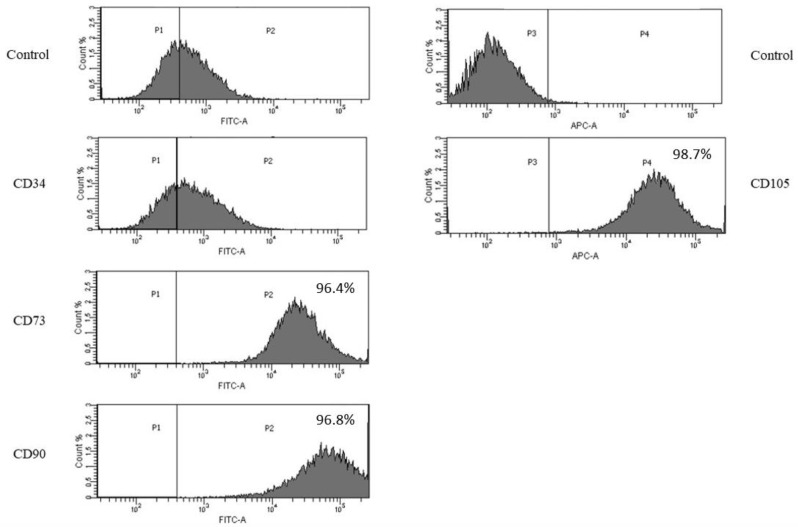
Flow cytometric analysis of MSC makers. Bone marrow MSCs were found to be positive for the MSC markers: CD73, CD90 and CD105 and negative for the endothelial marker: CD34.

**Figure 3 bioengineering-09-00754-f003:**
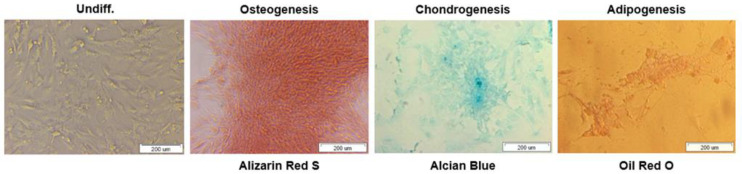
Differentiation of bmMSCs to osteocytes (Alizarin Red S), chondrocytes (Alcian Blue), and adipocytes (Oil Red O). Scale bars = 200 µm.

**Figure 4 bioengineering-09-00754-f004:**
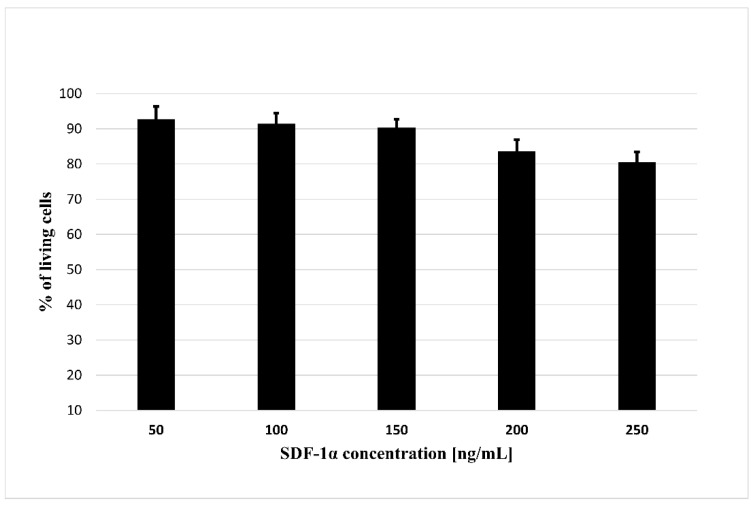
The cytotoxicity assay of the SDF-1α. The assay was performed with bmMSC cells. The results are presented as a% of cells surviving in the presence of the SDF-1α in the range of concentrations (50–250 ng/mL). The bars represent the means ± SD (*n* = 3).

**Figure 5 bioengineering-09-00754-f005:**
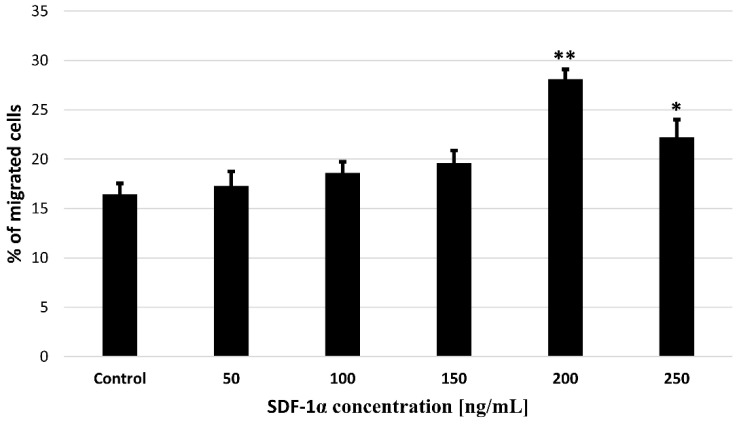
Effects of SDF-1α on bmMSCs migration in vitro. Quantification of migration of bmMSCs induced by different concentration of SDF-1α (10, 50, 100, 150, 200 and 250 ng/mL) for 24 h was detected with Transwell migration assay. Values are means ± SD (*n* = 10). *p*-values of statistical significance are represented as follows: * *p* < 0.05; ** *p* < 0.01.

**Figure 6 bioengineering-09-00754-f006:**
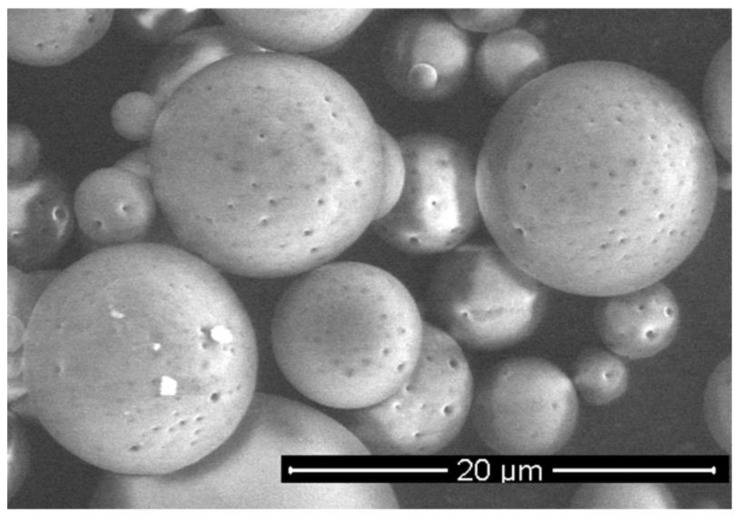
SEM image of SDF-1α-loaded MS.

**Figure 7 bioengineering-09-00754-f007:**
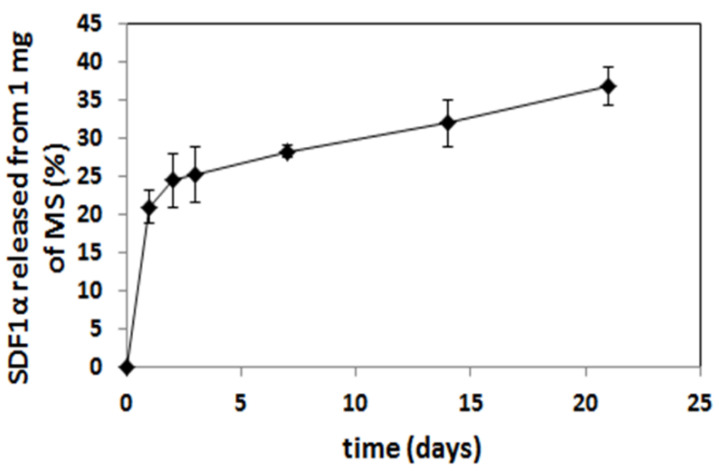
Cumulative in vitro release profile of SDF-1α from PLA/GA/TMC MS.

**Figure 8 bioengineering-09-00754-f008:**
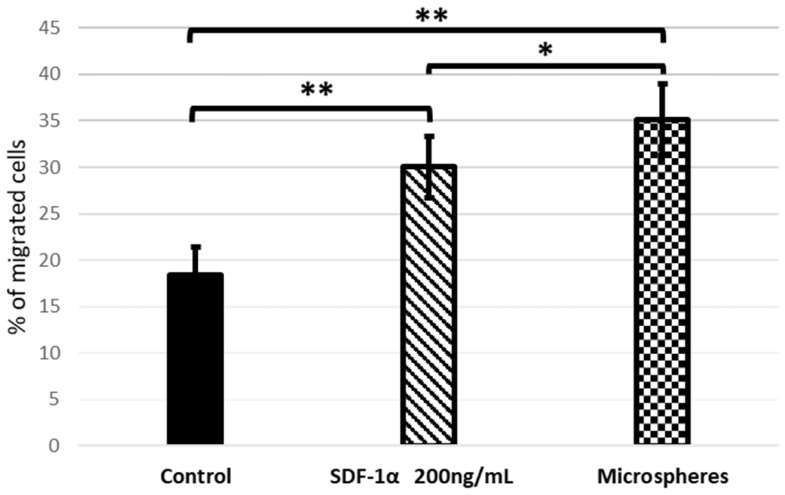
Transwell migration assay. Quantification of migration of bmMSCs after 24 h of treatment with SDF-1α and SDF-1α released from microspheres. Values are means ±SD (*n* = 10), assessed one-way ANOVA followed by Tukey’s post hoc test. *p*-values of statistical significance are represented as follows: * *p* < 0.05; ** *p* < 0.01.

## Data Availability

No new data were created or analyzed in this study. Data sharing is not applicable to this article.

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
