# Peer review of "Controlled Release of Encapsuled Stromal-Derived Factor 1α Improves Bone Marrow Mesenchymal Stromal Cells Migration"

_bioengineering, 2022, doi:10.3390/bioengineering9120754_

Round 1
Reviewer 1 Report
This paper evaluated the effects of SDF-1a-loaded microspheres on MSC migration. Overall, the study is straightforward and the methods are described well. Suggestions are made to improve the quality of the manuscript.
-Please provide quantification of bmMSC protein expression markers within the results (Fig 2).
-Please clarify in the Results how SDF-1a release was measured (immunoassay?) and temperature conditions.
-Given the high amount of SDF-1a detected at 1d (the first timepoint shown), how was it confirmed that this protein was encapsulated and not just surface-bound? Shorter timepoints (15 min) should also be included.
-It would be beneficial to include a migration study using later timepoints than just 24 hrs to show that the protein is released in a linear manner from the microspheres.
-Please correct grammatical and syntax issues throughout - ex: first sentence of abstract - 'quit' ; change 'thanks to' to 'attributed to' or 'as a function of' ; check superscripts for all values (106 should be '10^6) - line 102, 122; Fig. 1 change '2,5' to '2.5'; line 312 - modify phrase 'in regular way' to specify the quantitative change with time (e.g., x-fold reduction per day).
Author Response
Reviewer 1
General Comment: This paper evaluated the effects of SDF-1a-loaded microspheres on MSC migration. Overall, the study is straightforward and the methods are described well. Suggestions are made to improve the quality of the manuscript.
Response to General Comment: We very much appreciate the overall positive evaluation of our article.
Comment 1: Please provide quantification of bmMSC protein expression markers within the results (Fig 2).
Response to Comment 1: Values has been added onto histograms (Fig. 2).
Comment 2: Please clarify in the Results how SDF-1a release was measured (immunoassay?) and temperature conditions.
Response to Comment 2: The detailed procedure of SDF-1α release was described in the Section 2.11.3. In vitro release of the SDF-1α. As described in this section, the release analysis was conducted at 37 °C: “The suspension of 5mg MS with SDF-1α in 5mL of phosphate buffer saline (pH 7.4) was placed in 15mL tubes and incubated at 37°C under constant shaking of 240 rpm.“ and the immunoassay was used for measurement of SDF: ”The quantitative analysis of SDF-1α was conducted on the basis of the ELISA assay (Human SDF-1 ELISA Kit, AVIVA Systems Biology).”
However, according to the suggestion of the Reviewer 1, the additional information was also added in the Results section (3.5. Encapsulation and release of SDF-1α from MS): “The in vitro release of SDF-1α was conducted at 37 °C and the immunoenzymatic method was applied for quantitative evaluation.”
Comment 3: Given the high amount of SDF-1a detected at 1d (the first timepoint shown), how was it confirmed that this protein was encapsulated and not just surface-bound? Shorter timepoints (15 min) should also be included.
Response to Comment 3: The issue of the increased release of SDF-1α was explained in the discussion section: “Typically for microparticulate delivery systems, an increased initial SDF-1α release was observed after 1 day (21 %). This effect is explained by rapid liberation of proteins located adjacent to the microspheres’ surface [39]. The initial more rapid release may be useful in establishing higher local concentration of SDF-1α and providing immediate induction of chemotaxis of the stem cells. This stage was followed by a slower release caused probably by the diffusion of proteins from deeper parts of the microspheres.” It is considered that the accelerated initial drug release (burst effect) is caused by liberation of the drug located at or near the surface of the microspheres. The initial burst release is controlled by only diffusion, while the lag phase and secondary release are dependent on both diffusion and particle erosion [DOI: 10.3390/pharmaceutics13081313]. This phenomenon is typical for many types of delivery systems, including microspheres. Considering this fact, similar amount of drug release is expected after shorter timepoints, so it was not involved in the study. Generally, the microspheres were developed for long-term release, so the timepoints have been planned to provide information about the whole release period. Unfortunately, there are no clear recommendations or official method for drug release testing from polymeric microparticles, also for long-term release parenteral dosage forms [DOI 10.3390/pharmaceutics13081313; DOI https://doi.org/10.1007/s40005-013-0072-5; DOI: 10.1007/s11095-005-9397-8].
Comment 4: It would be beneficial to include a migration study using later timepoints than just 24 hrs to show that the protein is released in a linear manner from the microspheres.
Response to Comment 4: We are grateful for valuable comment. Migration assay in timepoints later than 25 hrs would be valuable for further use such microspheres in in vivo therapies. We have planned our experiments based on available articles. Previous experiments have shown that transwell migration assay has been successfully performed for B16F10 cells within 16 hrs [doi:10.3791/51046],for Neural Stem cells within 24 hrs [doi.org/10.1021/acsami.6b06780], B16 cells within 24 hrs [https://doi.org/10.1016/j.biopha.2020.109984]. We kindly ask for the possibility to keep the migration assay from its current form and present long-term analyses in the following works.
Comment 5: Please correct grammatical and syntax issues throughout - ex: first sentence of abstract - 'quit' ; change 'thanks to' to 'attributed to' or 'as a function of' ; check superscripts for all values (106 should be '10^6) - line 102, 122; Fig. 1 change '2,5' to '2.5'; line 312 - modify phrase 'in regular way' to specify the quantitative change with time (e.g., x-fold reduction per day).
Response to Comment 5: Changes have been added throughout the manuscript.
Reviewer 2 Report
The authors studied a very interesting topic to overpass the problem of cell engraftment and efficacy after cell transplantation. The authors propose a system that attracts endogenous cells to colonize specific sites in the body (e.g. the heart after ischemic injury). Those PLGA/TMC microspheres, inspired by an FDA-approved system, seem a relevant approach for drug delivery, especially its long-term release of the drug. The authors isolated mesenchymal stem cells and showed that specific doses of SDF1α are not toxic to cells. They chose 200ng/ml of SDF in microspheres as it was promoting the highest migration rate. The system SDF/microspheres have a more potent attracting effect on MSC than the control or SDF alone.
This is a very good study, and seems promising, but with the limitation that the authors did not confirm further yet with in vivo models.
- As a control, I would just suggest confirming that microspheres with SDF1α are non-toxic also for adult cardiomyocytes, as for their application into the heart.
Author Response
Reviewer 2
General Comment: The authors studied a very interesting topic to overpass the problem of cell engraftment and efficacy after cell transplantation. The authors propose a system that attracts endogenous cells to colonize specific sites in the body (e.g. the heart after ischemic injury). Those PLGA/TMC microspheres, inspired by an FDA-approved system, seem a relevant approach for drug delivery, especially its long-term release of the drug. The authors isolated mesenchymal stem cells and showed that specific doses of SDF1α are not toxic to cells. They chose 200ng/ml of SDF in microspheres as it was promoting the highest migration rate. The system SDF/microspheres have a more potent attracting effect on MSC than the control or SDF alone. This is a very good study, and seems promising, but with the limitation that the authors did not confirm further yet with in vivo models.
Response to General Comment: We very much appreciate the overall positive evaluation of our review. We are grateful for valuable comments.
Comment 1: As a control, I would just suggest confirming that microspheres with SDF1α are non-toxic also for adult cardiomyocytes, as for their application into the heart.
Response to Comment 1: We have already assessed the application of microspheres in in vivo models. This findings will be published soon. Similarly, cytotoxicity of microspheres with SDF1α for wide range of cells will be also assessed and published.
Reviewer 3 Report
The research team proposed the newly created biodegradable polymeric microspheres for the release of stromal derived factor SDF-1alpha in a controlled and long-lasting manner for further application in stem cells therapy. The manuscript is well-written and detailed; the research topic is supported with experimental results and arguments, well explained and summarized.
I would kindly recommend just a minor spell-check of the text. Below are some examples of what should be corrected. Thank you very much and lots of success for the research team with the publication.
Line 251: MSC markers
Line 349: therapies for the treatment of
Line 441: It will allow (for) the controlled delivery
Please check throughout the text, phrases like in vivo, in vitro - should be in italics.
Author Response
Reviewer 3
General Comment: The research team proposed the newly created biodegradable polymeric microspheres for the release of stromal derived factor SDF-1alpha in a controlled and long-lasting manner for further application in stem cells therapy. The manuscript is well-written and detailed; the research topic is supported with experimental results and arguments, well explained and summarized.
Response to General Comment: We very much appreciate the overall positive evaluation of our review. We are grateful for valuable comments. I would kindly recommend just a minor spell-check of the text. Below are some examples of what should be corrected. Thank you very much and lots of success for the research team with the publication.
Comment 1: Line 251: MSC markers
Response to Comment 1: This has been corrected accordingly.
Comment 2: Line 349: therapies for the treatment of
Response to Comment 2: This has been corrected accordingly.
Comment 3: Line 441: It will allow (for) the controlled delivery
Response to Comment 3: This has been corrected accordingly.
Comment 4: Please check throughout the text, phrases like in vivo, in vitro - should be in italics.
Response to Comment 4: Changes have been added throughout the manuscript.
Round 2
Reviewer 2 Report
Thank you very much for the information. My concern was about the safety of the TMC part of the microspheres. Good luck with the in vivo studies. The manuscript is suitable for publication.